# Gaussian Process Latent Variable Model: A Performance Upper Bound with Noise-Corrupted Data

## Abstract

Real-world datasets are often corrupted with noise. Probabilistic models are developed for learning in such scenarios, particularly where data samples are noisy and uncertainty needs to be considered, due to their probabilistic inference framework. Various types of probabilistic models, such as the Bayesian Gaussian Process Latent Variable Model (BGPLVM), are widely used in learning problems that emphasize uncertainty. However, despite the promising results from these probabilistic models, an analytic performance analysis with noise-corrupted uncertain data has not yet been conducted. In this paper, we focus on the BGPLVM and propose to analyze the performance upper bound of probabilistic models for clustering tasks quantitatively. We review the BGPLVM and propose an analytic performance upper bound, defined as the minimum probability of false alarm for clustering problems with datasets corrupted by Gaussian noise. This upper bound represents the best performance achievable by any clustering algorithm, regardless of the specific algorithm used, with BGPLVM serving as the means of dimensionality reduction. The results derived from Gaussian noise scenarios are then generalized to non-Gaussian scenarios. Numerical results are provided to validate our proposed performance upper bound of the BGPLVM in clustering tasks with noise-corrupted data. This framework can be generalized to evaluate the performance upper bound of a wide class of probabilistic models.

## 1 Introduction

Probabilistic models are a cornerstone of machine learning, offering a structured approach to dealing with uncertainty and variability in data. These models use probability distributions to describe the inherent uncertainties in data and model parameters, allowing for more flexible and robust inferences. By accounting for noise and uncertainty, probabilistic models can provide more reliable predictions and insights compared to deterministic models. They are particularly useful in real-world applications where data is often noisy, incomplete, or ambiguous. Examples of probabilistic models include Bayesian networks (Friedman et al., 1997; Kitson et al., 2023; Margaritis et al., 2003), hidden Markov models (Rabiner & Juang, 1986; Fine et al., 1998; Eddy, 2004), and Gaussian processes (Williams & Rasmussen, 1995; McHutchon & Rasmussen, 2011; Damianou & Lawrence, 2013; Liu et al., 2020; Wilson et al., 2020), each of which applies probabilistic reasoning to different types of data and learning tasks.

Among the prevailing probabilistic models, Gaussian Processes (GPs) are a powerful type of probabilistic model defined by a mean function and a covariance function, which together describe a distribution over possible functions that fit the observed data. This allows GPs to provide not just predictions but also uncertainty estimates for those predictions, making them particularly valuable in scenarios where understanding the confidence in the predictions is important. The Bayesian Gaussian Process Latent Variable Model (BGPLVM) (Titsias & Lawrence, 2010; de Souza et al., 2021; Gundersen et al., 2021; Lalchand et al., 2022), an extension of GP, inherits the advantages of the original GP while introducing additional capabilities. BGPLVM is a multiple-output GP regression model where only the output data are given, and the inputs are unobserved latent variables. This model can be applied to clustering and other unsupervised learning tasks. In BGPLVM, the latent variables are not integrated out but are instead optimized during the learning process. This approach makes the model more tractable and allows for efficient inference. BGPLVM

introduces a variational inference framework to train the model, facilitating Bayesian nonlinear dimensionality reduction. This framework approximates the posterior distribution of the latent variables and model parameters, making it possible to handle high-dimensional data and complex probabilistic structures.

In real-world datasets, the obtained data are often corrupted by noise. This can occur due to inaccuracies in human data annotation, sensor noise during data collection, and other factors. Learning from noise-corrupted datasets is a significant challenge and an important area of research (Li et al., 2019a; 2021; Niyogi et al., 2011). Additionally, safe learning, a prominent research topic, focuses on developing algorithms that not only perform well but also provide guarantees or quantifiable metrics regarding their performance, robustness, and reliability when dealing with noisy data samples (Li et al., 2019b; Wu et al., 2021; Lütjens et al., 2019).

Several researchers have focused on developing novel Gaussian Process models to enhance the performance of regression and clustering tasks with noise-corrupted data (Goldberg et al., 1997; Villacampa-Calvo et al., 2021; Liu et al., 2022). The proposed algorithms have demonstrated promising performance on various noise-corrupted datasets. However, to our knowledge, an analytic performance analysis of using Gaussian process models for clustering problems has not been conducted. Quantitative error analyses for learning algorithms with noise-corrupted datasets are also rare. Although deriving performance bounds for complex machine learning models is a challenging task (Lui et al., 2018), the probabilistic inference structure of Gaussian process models makes it feasible to derive performance bounds. However, this has not yet been addressed in the literature.

The paper is organized as follows. First, we provide a brief review of the Bayesian Gaussian Process Latent Variable Model (BGPLVM), with a focus on its evidence lower bound. Next, we analyze the effect of additive noise in the observations on the variational posterior distribution in the latent space. We then formulate the clustering problem as a multiple hypothesis testing problem and use the Chernoff-Stein Lemma to relate the KL divergence in the latent space to the probability of false alarm. We derive a performance upper bound, defined as the minimum probability of false alarm ($P_F$), for noise-corrupted data clustering using BGPLVM. This bound is independent of specific clustering algorithms, making it broadly applicable to real-world scenarios. Finally, we provide simulation results on a Single-Cell qPCR dataset to verify the theoretical bound.

## 2 Bayesian Gaussian Process Latent Variable Model

Following the definition of BGPLVM in (McHutchon & Rasmussen, 2011), let $Y \in \mathbb{R}^{N \times D}$ be the observation data set where $N$ is the number of observations and $D$ is the dimensionality of each data vector. And we assume the data are associated with latent variables $X \in \mathbb{R}^{N \times Q}$. And as BGPLVM is used for dimension reduction, we have $Q \ll D$. The BGPLVM defines a generative mapping from the latent space to observation space which is governed by Gaussian processes. If the GPs are taken to be independent across the features then the likelihood function is written as

$$p(Y \mid X) = \prod_{d=1}^{D} p(y_d \mid X) \tag{1}$$

$$p(y_d \mid X) = \mathcal{N}\left(y_d \mid 0, K_{NN} + \beta^{-1}I_N\right) \tag{2}$$

where $y_d$ represents the $d^{\text{th}}$ column of $Y$. $K_{NN}$ is the $N \times N$ covariance matrix defined by the kernel function $k(x, x')$. equation 1 gives the likelihood function of a multiple-output GP regression model where the vectors of different outputs are drawn independently from the same Gaussian process prior which is evaluated at the inputs $X$. Since $X$ is a latent variable, a prior density given by the standard normal density is assigned

$$p(X) = \prod_{n=1}^{N} \mathcal{N}\left(x_n \mid 0, I_Q\right)$$

where each $x_n$ is the $n^{\text{th}}$ row of $X$. Now the joint probability model for the BGPLVM model is

$$p(Y, X) = p(Y \mid X)p(X).$$

The marginal likelihood of the data

$$p(Y) = \int p(Y \mid X)p(X)dX.$$

But the quantity is intractable as $X$ appears non-linearly inside the inverse of the covariance matrix $K_{NN} + \beta^{-1}I_N$. Thus we seek to apply an approximate variational inference procedure where an approximate of the true posterior distribution, i.e. $q(X)$ is introduced to approximate the true posterior distribution $p(X \mid Y)$ over the latent variables,

$$q(X) = \prod_{n=1}^{N} \mathcal{N}\left(x_n \mid \mu_n, S_n\right) \tag{3}$$

where the variational parameters are $\{\mu_n, S_n\}_{n=1}^{N}$. Here for simplicity, $S_n$ is taken to be a diagonal covariance matrix. By using this variational distribution we can express an evidence lower bound on the $\log p(Y)$ that takes the form (Titsias & Lawrence, 2010)

$$
\begin{aligned}
&F(q) \\
&= \int q(X) \log \frac{p(Y \mid X)p(X)}{q(X)} dX \\
&= \int q(X) \log p(Y \mid X)dX - \int q(X) \log \frac{q(X)}{p(X)} dX \\
&= \tilde{F}(q) - KL(q\|p)
\end{aligned}
$$

where the second term is the KL divergence between the variational posterior distribution $q(X)$ and the prior distribution $p(X, Y)$. And the goal of BGPLVM is to obtain $\max F(q)$.

In real-world datasets, the observations are always corrupted with noise. In the following theorem, we first look into the minimum KL divergence with additive noise for the BGPLVM.

**Lemma 1** (Additive Noise (Polyanskiy & Wu, 2014))**.**

$$
\begin{aligned}
&Y = X + Z, Z \perp X \\
&\Leftrightarrow p(Y \mid X = x) = p(x + Z) \\
&\Leftrightarrow p(Y = y, X = x) = p(x + Z)p(X = x)
\end{aligned}
$$

*where $Z$ is an independent additive noise.*

**Theorem 2** (Minimum KLD with additive Gaussian noise)**.** *For training data corrupted with additive Gaussian noise $Z$, the noise-corrupted observation can be denoted as $Y = \underline{Y} + Z$, where $\underline{Y}$ is the noise-free data, and $Z \sim \mathcal{N}\left(0, \sigma^2\right)$. Then $F_N\left(q_N\right)$ is maximized when the variational posterior distribution with noise $q_N(X, Z) = q(X)p(\underline{Y} + Z)$.*

*Proof.* By Lemma 1, we have

$$p_N(Y, Z \mid X) = p(Y = \underline{Y} \mid X)p(\underline{Y} + Z),$$

where the true latent variables $X$ are not affected by the noise $Z$, making them independent. The evidence lower bound with noise data can then be written as

$$
\begin{aligned}
&F_N\left(q_N\right) \\
&= \iint q_N(X, Z) \log \frac{p_N(Y, Z \mid X)p(X)}{q_N(X, Z)} dXdZ \\
&= \iint q_N(X, Z) \log \frac{p(Y = \underline{Y} \mid X)p(\underline{Y} + Z)p(X)}{q_N(X, Z)} dXdZ
\end{aligned}
$$

where $q_N(X, Z)$ corresponds to the approximate variational posterior distribution with noise corrupted data. By (Polyanskiy & Wu, 2014)[Corollary 2.1], we further have

$$\arg \max_{q_N(X, Z)} F_N(q_N) = q(X)q(Z).$$

Then we have

$$F_N\left(q_N(X, Z)\right)$$
$$\leq \tilde{F}_N(q(X)) - KL(q(X)\|p(X)) - KL(q(Z)\|p(\underline{Y} + Z))$$

with equality achieved if and only if $q_N(X, Z) = q(X)q(Z)$.

Because $KL(q(Z)\|p(\underline{Y} + Z)) \geq 0$, then we have

$$\max F_N\left(q_N(X, Z)\right) =$$
$$\tilde{F}_N(q(X)) - KL(q(Z)\|p(\underline{Y} + Z)) \quad (4)$$
$$\iff p(\underline{Y} + Z) = q(Z).$$

$\square$

By Theorem 2, we can conclude that additive Gaussian noise in the observation space is captured by the variational posterior distribution in the latent space.

## 3 Clustering Performance and KL Divergence

In the previous section, we analyzed the relationship between the KL divergences and the noise of the observation data. As our task is to investigate the performance of clustering using BGPLVM with noise corrupted data, we first propose a metric to characterize the performance.

### 3.1 Probability of False Alarm and Clustering

The probability of false alarm is associated with Type I error in hypothesis testing, i.e., the probability of falsely rejecting the null hypothesis. Meanwhile, clustering problems can be regarded as multiple hypothesis testing.

Assume a set $\mathbf{C}$ which are the true categories of all data samples. Denote the $M$ subsets of the ground-truth categories as $\{\mathbf{C}_i\}_{i=1}^M$. Then the clustering problem can be considered as the problem of simultaneously testing a null hypothesis $H_0^i$ against the alternative hypothesis $H_1^i$, for $i = 1, \ldots, M$. The null hypothesis and alternative hypothesis can be written as:

$$H_0^i : \text{observation } Y^* \notin \mathbf{C}_i \quad H_1^i : \text{observation } Y^* \in \mathbf{C}_i.$$

If we further assume that each data sample has a unique category, only one alternative hypothesis $H_1^i$ will be accepted. In this regard, we could regard the clustering algorithm as a detector. Denoting the clustering results using BGPLVM and an arbitrary detection algorithm, by the hypothesis testing as $\hat{\mathbf{C}}_i$, the hypothesis testing problem can then be formulated as

$$H_0 : \text{observation } Y^* \notin \hat{\mathbf{C}}_i \quad H_1 : \text{observation } Y^* \in \hat{\mathbf{C}}_i.$$

Then a false alarm occurs when $H_0$ is rejected but $Y^* \notin \hat{\mathbf{C}}_i$. In the setting of this paper, it is equivalent to $\hat{\mathbf{C}}_i \neq \mathbf{C}_i$.

For the clustering problems, we are interested in the error (misclustering) rate (Dalmaijer et al., 2022). The error rate can be written as a sum of probability of false alarm of each category, i.e. $P_F = \sum_{i=1}^M P_F^i$, if each observed point is classified into a category.

Now the clustering problem has been written in a hypothesis testing manner. In the Chernoff-Stein lemma, the relationship between KL divergence and the probability of false alarm has been proved. This lemma enables us to analyze the performance of a clustering (detection) problem, in the sense of the probability of false alarm, by the KL divergence.

### 3.2 Probability of False Alarm and KL Divergence

As to relate the KL divergence to the probability of false alarm, we first review the Chernoff-Stein Lemma.

**Lemma 3** (Chernoff-Stein Lemma (Cover, 1999)). *Let $Y_1, Y_2, \ldots, Y_N$ be i.i.d. samples drawn from a distribution $Q \in \mathcal{P}(\mathcal{Y})$. Consider the hypothesis test with the null hypothesis*

$$H_0 : Q = P_0$$

*and the alternative hypothesis*

$$H_1 : Q = P_1.$$

*Assume that the Kullback-Leibler divergence $KL(P_0 \| P_1)$ is finite.*

*Define $\mathcal{A}_N \subset \mathcal{Y}^N$ as the acceptance region for the null hypothesis $H_0$. The probabilities of error are given by:*

$$P_M^N(\mathcal{A}_N) \doteq P_0^N(\mathcal{A}_N^c) \quad \text{(Type I error)}$$

*and*

$$P_F^N(\mathcal{A}_N) \doteq P_1^N(\mathcal{A}_N) \quad \text{(Type II error)}.$$

*For a given $\epsilon \in (0, 1)$, define $P_F^{N*}(\epsilon)$ as:*

$$P_F^{N*}(\epsilon) \doteq \min \left\{ P_F^N(\mathcal{A}_N) : P_M^N(\mathcal{A}_N) \leq \epsilon \right\},$$

*where $\mathcal{A}_N$ ranges over all subsets of $\mathcal{Y}^N$.*

*The Chernoff-Stein lemma states that for every $\epsilon \in (0, 1)$,*

$$\lim_{N \to \infty} -\frac{1}{N} \log P_F^{N*}(\epsilon) = KL(P_0 \| P_1).$$

*Remark.* Lemma 3 reveals the fact that regardless of the constant upper bound on the type-I error, the type-II error always behaves as $P_F^{N*}(\epsilon) = 2^{-NKL(P_0\|P_1)}$, with a relatively large $N$. This provides an operational meaning of the relative entropy $KL(P_0\|P_1)$ as the best exponent for discriminating the two distributions $P_0$ and $P_1$. However, when an arbitrary clustering algorithm is applied to the latent space obtained by the BGPLVM and the probability of false alarm is denoted as $P_F^N$, we have

$$\lim_{N \to \infty} -\frac{1}{N} \log P_F^N(\epsilon) \leq \lim_{N \to \infty} -\frac{1}{N} \log P_F^{N*}(\epsilon) = KL(P_0 \| P_1).$$

In real-world datasets, the dimension of the data sample is always limited, i.e. $N < \infty$, we have

$$-\frac{1}{N} \log P_F^N(\epsilon) \leq \lim_{N \to \infty} -\frac{1}{N} \log P_F^N(\epsilon).$$

Lemma 3 reveals the relationship between the KL divergence and the probability of false alarm. In the following part of this section, we apply it to observe the probability of false alarm for the BGPLVM. We denote the observation data corrupted by a bias as $\breve{Y} = \underline{Y} + b$, where $\breve{}$ denotes the bias. To examine performance under noise, the observation corrupted with additive noise is denoted as $\breve{Y}_1 = \breve{Y} + Z_1$. The observation corrupted solely with additive noise $Z_2$ is denoted as $Y_2 = \underline{Y} + Z_2$. The posterior distributions of the two types of observations approximated by BGPLVM are denoted as $q(X \mid b, Z_1)$ and $q(X \mid 0, Z_2)$. Then, the probability of false alarm for BGPLVM is derived in the following lemma.

**Lemma 4** (Probability of False Alarm of BGPLVM for Clustering). *Denote the probability of a false alarm for clustering using BGPLVM as $\breve{P}_F^N$ (with bias and noise) and $P_F^N$ (with noise only) respectively, then*

$$-\frac{1}{N} \log \breve{P}_F^N = l \left( KL \left( q \left( X \mid b, Z_1 \right) \| q \left( X \mid 0, Z_2 \right) \right) \right),$$

$$-\frac{1}{N} \log P_F^N = l \left( KL \left( q \left( X \mid 0, Z_1 \right) \| q \left( X \mid 0, Z_2 \right) \right) \right),$$

*where $0 \leq l(x) \leq x$ and $\frac{l(x_1)}{x_1} \leq \frac{l(x_2)}{x_2}$, given $0 \leq x_1 \leq x_2$.*

*Proof.* By Lemma 3, we have

$$-\frac{1}{N} \log \breve{P}_F^N \leq \lim_{N \to \infty} -\frac{1}{N} \log \breve{P}_F^N \leq \mathrm{KL}\left(q\left(X \mid b, Z_1\right) \| q\left(X \mid 0, Z_2\right)\right),$$

and

$$-\frac{1}{N} \log P_F^N \leq \lim_{N \to \infty} -\frac{1}{N} \log P_F^N$$
$$\leq \mathrm{KL}\left(q\left(X \mid 0, Z_1\right) \| q\left(X \mid 0, Z_2\right)\right).$$

We note that the same detector, represented by $l(\cdot)$, is used in the clustering problem. The condition $\frac{l(x_1)}{x_1} \leq \frac{l(x_2)}{x_2}$ holds because, as the KL divergence increases, the detection problem with additive noise becomes easier and its performance approaches that of the problem without noise. An illustrative example of $l(\cdot)$ is given in Figure 5 in Appendix A. □

### 3.3  Calculation of $P_F$

In this subsection, we calculate $P_F$ to give an analytic solution to the performance upper bound of the clustering using BGPLVM. By Theorem 2, we can obtain

$$KL\left(q\left(X \mid b, Z_1\right) \| q\left(X \mid 0, Z_2\right)\right) = KL(q(X \mid b, 0) \| q(X \mid 0, 0)) + KL\left(q\left(Z_1\right) \| q\left(Z_2\right)\right). \tag{5}$$

equation 5 reveals the fact that the KL divergence term can be segmented into two terms, one is the KL divergence of difference in the latent space and the other one is the KL divergence of the noises.

So far, we have derived the KL divergences in the BGPLVM with additive Gaussian noise. We will then proceed to calculate the probability of false alarm, $P_F$, using the derived KL divergences.

**Lemma 5** (Probabilities of false alarm without difference).

$$\frac{\log \breve{P}_F^N}{\log P_F^N} \leq \frac{KL\left(q\left(X \mid b, Z_1\right) \| q\left(X \mid 0, Z_2\right)\right)}{KL\left(q\left(X \mid 0, Z_1\right) \| q\left(X \mid 0, Z_2\right)\right)} \tag{6}$$

*Proof.* By Theorem 3, we obtain

$$\frac{\log \breve{P}_F^N}{\log P_F^N} = \frac{l\left(KL\left(q\left(X \mid b, Z_1\right) \| q\left(X \mid 0, Z_2\right)\right)\right)}{l\left(KL\left(q\left(X \mid 0, Z_1\right) \| q\left(X \mid 0, Z_2\right)\right)\right)}. \tag{7}$$

The proof then follows directly from the Law of Sines. □

**Theorem 6** (Performance Upper Bound $P_F$ Corrupted with Additive Noise $Z_1$).

$$\log \breve{P}_F^N$$
$$\geq \left(1 + \frac{KL(q(X \mid b, 0) \| q(X \mid 0, 0))}{KL\left(q(Z_1) \| q(Z_2)\right)}\right) \cdot \log P_F^N \tag{8}$$

*Proof.* By equation 6 and Theorem 2, and due to $\log P_F < 0$, we have equation 9, which completes the proof. □

We have now obtained the least probability of false alarm, which is the performance upper bound of BGPLVM for clustering with noise-corrupted data.

By Theorem 3.3, we observe that as the difference between the observation and the trained model increases, the lower bound of $P_F$ decreases. Conversely, as the additive Gaussian noise increases, the lower bound of $P_F$ increases. This reveals how the additive Gaussian noise affects the performance of BGPLVM in clustering noise-corrupted data.

$$\log \breve{P}_F^N \geq \frac{KL\left(q\left(X \mid b, Z_1\right) \| q\left(X \mid 0, Z_2\right)\right)}{KL\left(q\left(X \mid 0, Z_1\right) \| q\left(X \mid 0, Z_2\right)\right)} \log P_F^N = \frac{KL(q(X \mid b, 0) \| q(X \mid 0, 0)) + KL\left(q(Z_1) \| q(Z_2)\right)}{KL\left(q(Z_1) \| q(Z_2)\right)} \log P_F^N$$

$$= \left(1 + \frac{KL(q(X \mid b, 0) \| q(X \mid 0, 0))}{KL\left(q(Z_1) \| q(Z_2)\right)}\right) \log P_F^N. \tag{9}$$

### 3.4 A Generalization to Non-Gaussian Noises

For the results in the previous sections, we have mainly focused on the performance upper bound analysis of the Gaussian noises. However, the analysis in this paper could be generalized to non-Gaussian noises.

Since $p(\underline{Y} + Z_1)$ is not Gaussian, equation 4 is not valid any longer, where $q(Z_1)$ is Gaussian in the BGPLVM. Under this circumstance, the noise learned by the BGPLVM in the latent space is the Gaussian distribution which is closest to the non-Gaussian distribution, namely

$$q(Z_1) = \arg \min_{q(Z_1) \in \mathcal{N}} KL(q(Z_1) \| p(\underline{Y} + Z_1)). \tag{10}$$

where $\mathcal{N}$ denotes the set of all Gaussian distributions. The performance upper bound equation 8 is then calculated by $q(Z_1)$ in equation 10.

## 4 Numerical Results

In the previous sections, we have proposed a performance upper bound of the clustering task with noise-corrupted data samples, when BGPLVM and an arbitrary clustering algorithm in low dimension is used. To demonstrate the performance upper bound, in the sense of probability of false alarm, when BGPLVM is utilized for dimension reduction in real clustering tasks, we consider the clustering of cells using the Single-Cell quantitative polymerase chain reaction (qPCR) Dataset (Guo et al., 2010).

Single-cell qPCR (Isakova et al., 2021) is a powerful technique used to measure gene expression at the level of individual cells. This method provides a high-resolution view of cellular heterogeneity, which is crucial for understanding various biological processes, such as development, disease progression, and response to treatments. However, the noises are prevalent in Single-cell qPCR, due to the following reasons. Firstly, single cells inherently exhibit biological variability due to differences in gene expression levels, cell cycle stages, and metabolic states. Moreover, isolating and handling single cells is technically challenging and can introduce errors, such as incomplete cell lysis or mRNA degradation. The sensitivity and inaccuracies of the qPCR instrument in detecting fluorescence are not ignorable either, leading to noises in the measurement.

In Theorem 6, an upper bound of performance for clustering using the BGPLVM, in the sense of the probability of false alarm, is derived. However, in general, it is not feasible to directly obtain $KL(q(X \mid b, 0) \| q(X \mid 0, 0))$. Indeed, the integral can be approximated by Monte-Carlo approximation. For each simulation $i$ in $I$ simulations in total, we draw $2M$ samples from the whole dataset and separate them into two subsets, each with $M$ samples. Train the two subsets by BGPLVM, and obtain the corresponding variational posterior distribution as $q(X | b_{\mathcal{M}_1, i}, 0)$ and $q(X | b_{\mathcal{M}_2, i}, 0)$. Then by equation 3 we have

$$KL(q(X \mid b, 0) \| q(X \mid 0, 0)) = \int q(X \mid b, 0) \log \frac{q(X \mid b, 0)}{q(X \mid 0, 0)} dX$$

$$\approx \frac{N}{M \cdot I} \sum_{i=1}^{I} \int q(X | b_{\mathcal{M}_1, i}, 0) \log \frac{q(X | b_{\mathcal{M}_1, i}, 0)}{q(X | b_{\mathcal{M}_2, i}, 0)} dX. \tag{11}$$

However, calculating $KL(q(X \mid b, 0) \| q(X \mid 0, 0))$ can be complicated, especially when the number of data samples is large in the dataset. Since our training data is labeled, practically it is more convenient for us to calculate $KL(q(X \mid b, 0) \| q(X \mid 0, 0))$ by the probabilities of false alarm by simulation with small noises

added to the training data. $\sigma_2^2$ in equation 8 could be given prior (e.g. by the specifications of the sensor). If not available, we can pick a small value of $\sigma_2^2$. Then this value could be approximated by

$$\frac{1 + \frac{KL(q(X|b,0)\|q(X|0,0))}{KL\left(q(\hat{\sigma}_{11}^2)\|q(Z_2)\right)}}{1 + \frac{KL(q(X|b,0)\|q(X|0,0))}{KL\left(q(\hat{\sigma}_{12}^2)\|q(Z_2)\right)}} \approx \frac{\log \hat{P}_F(\hat{\sigma}_{11})}{\log \hat{P}_F(\hat{\sigma}_{12})}, \tag{12}$$

where $\hat{\sigma}_{11}^2$ and $\hat{\sigma}_{12}^2$ are two small positive values of the variances. $KL\left(q(\hat{\sigma}_{11}^2)\|q(Z_2)\right)$ and $KL\left(q(\hat{\sigma}_{12}^2)\|q(Z_2)\right)$ denote the $KL\left(q(Z_1)\|q(Z_2)\right)$ with respect to $Z_1 \sim \mathcal{N}(0, \hat{\sigma}_{11}^2)$ and $Z_1 \sim \mathcal{N}(0, \hat{\sigma}_{12}^2)$ correspondingly. $\hat{P}_F$ here denotes the probability of false alarm by numerical simulations, which is to distinguish it from $P_F$, the theoretical lower bound of the probability of false alarm. $\hat{P}_F(\hat{\sigma}_{11})$ and $\hat{P}_F(\hat{\sigma}_{12})$ correspond to $Z_1 \sim \mathcal{N}(0, \hat{\sigma}_{11}^2)$ and $Z_1 \sim \mathcal{N}(0, \hat{\sigma}_{12}^2)$ respectively. Details of the approximation is given in Appendix A.

This dataset consists of single-cell qPCR data for 48 genes obtained from mice, which is available at the Open Data Science repository. The data fall within 10 catogories. We generate the additive Gaussian noise sequence with $\sigma_1$ ranging from 0.1 to 0.6, and add it to the raw training data samples. We choose $\hat{\sigma}_{11}^2 = 0.25^2$ and $\hat{\sigma}_{12}^2 = 0.35^2$ in this simulation. The simulation results show that when $\sigma_2^2$ is small enough, the increase of lower bound of $P_F$ with the decrease of $\sigma_2^2$ is very small. The number of samples with the label of one class classified mistakenly to another class (denoted $N_F$) is obtained with different $\sigma_1^2$. Then $P_F = \frac{N_F}{N}$ is the probability of false alarm from experiment. Optimal performance lower bound of $P_F$ is calculated for comparison.

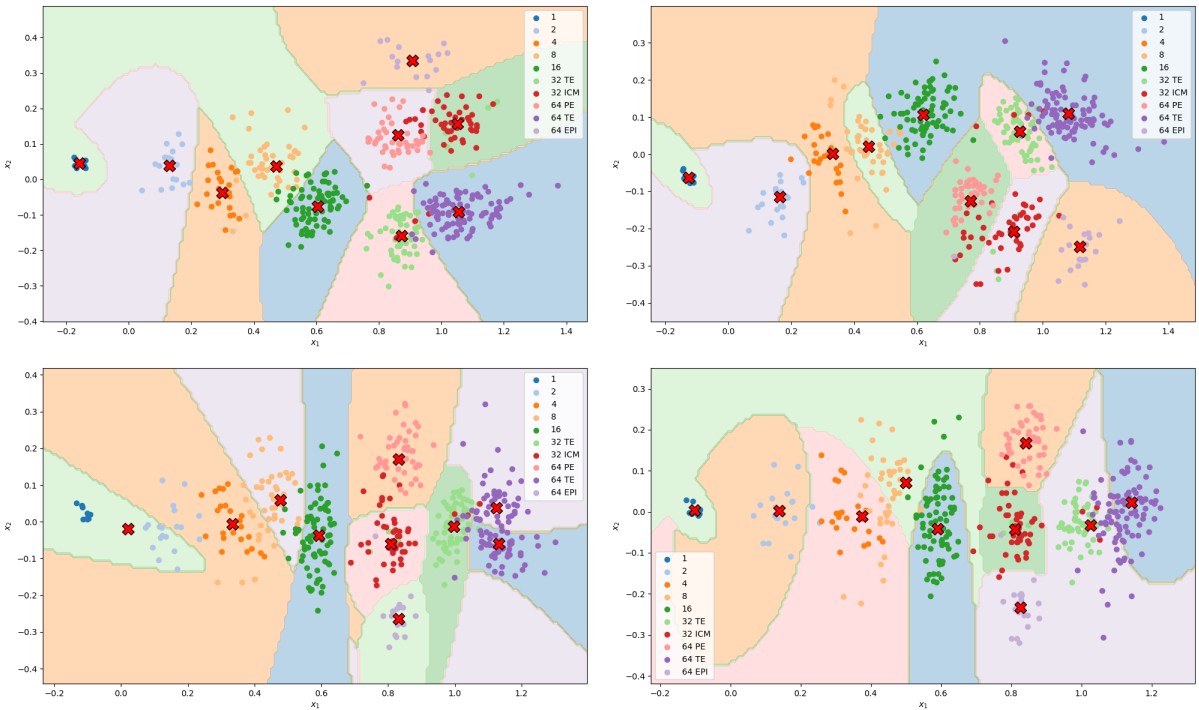

Figure 1: Simulation results of clustering the noise-corrupted single-cell qPCR data using a GMM, with dimension reduction performed by BGPLVM. The upper left, upper right, lower left, lower right subfigures correspond to the data without noise, and with noise at standard deviations of $\sigma = 0.2, 0.4, 0.6$ respectively.

Figure 1 shows the clustering results of the dimension-reduced data samples in the latent space obtained via BGPLVM, utilizing the Gaussian Mixture Model (GMM) with the standard deviation of the additive noise being $\sigma_1 = 0, 0.2, 0.4, 0.6$. Figure 2 provides a comparison between the estimated false alarm probability $\hat{P}_F$ derived from numerical simulations and the theoretical false alarm probability $P_F$ calculated based on our

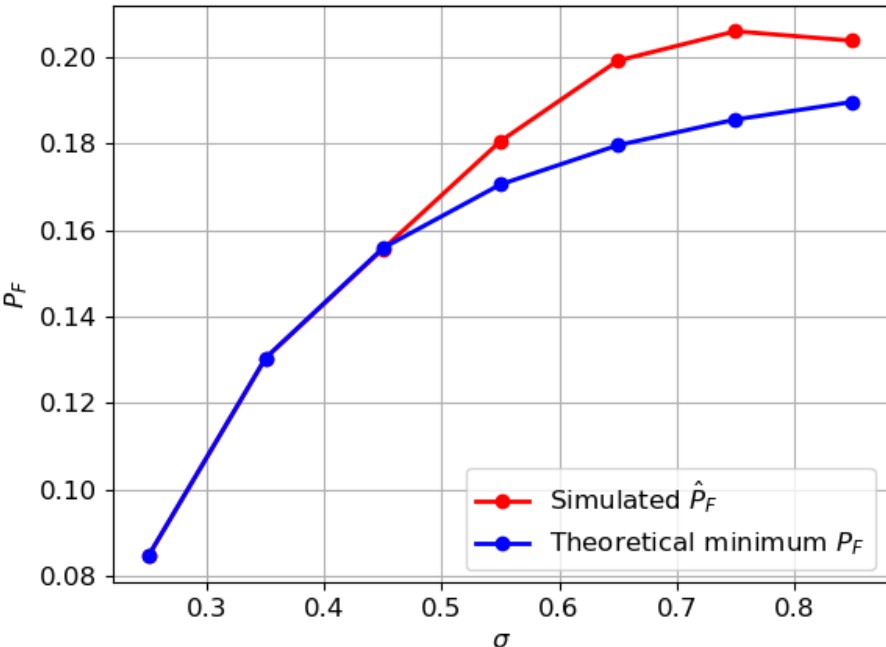

Figure 2: The probability of false alarm obtained from the simulation results and the proposed theoretical minimum probability of false alarm as a function of the standard deviation of the additive noise.

proposed method. The simulation results validate our performance upper bound as the minimum probability of false alarm.

In the simulations, we observe that when the additive Gaussian noise is small, the false alarm probability $p_F$ obtained from the simulation is close to the theoretical bound. However, when the noise variance increases, the clustering results exhibit a larger $\hat{p}_F$ compared to the theoretical bound. This discrepancy can be attributed partly to the relatively small dimensionality of the data (48 dimensions). Additionally, the theoretical bound is derived under the assumption that the number of data samples approaches infinity, whereas our simulation is conducted with a finite sample size of 437 data samples.

The proposed theoretical bound provides a quantifiable performance upper bound, representing the first quantitative safety measure for noise-corrupted data in clustering tasks, to the best of our knowledge. This bound offers the best analytic performance under certain noise levels, which is highly significant for safe learning applications where safety and reliability are crucial. Furthermore, the implications of this paper extend beyond the clustering problem itself, as the results can be generalized to a wide class of probabilistic models. This work underscores the significant potential of probabilistic models in safe learning tasks.

We have present a comparison of the performance upper bound for clustering the Single-cell qPCR dataset using the BGPLVM with the simulation results. The simulation results demonstrate that the simulated $\hat{P}_F$ consistently exceeds the theoretical minimum $P_F$ across all standard deviations of the additive Gaussian noise. As discussed in Section 3, we assert that this performance upper bound can also be extended to non-Gaussian noises, significantly enhancing the practical value of this bound in real engineering scenarios. In this section, we will put forward simulations on clustering of the Single-cell qPCR dataset using the BGPLVM and the GMM. Simulation settings follow those in Section 4, except for that the additive noise $Z_1$ is a Rayleigh distribution

$$Z_1 \sim \mathcal{R}(\sigma) = \frac{x}{\sigma^2} e^{-x^2/(2\sigma^2)}, \quad x \geq 0,$$

of which the mean is $\sigma\sqrt{\frac{\pi}{2}}$ and the variance is $\sigma^2\left(2 - \frac{\pi}{2}\right)$. By equation 10, we have that

$$q(Z_1) = \mathcal{N}\left(\sigma\sqrt{\frac{\pi}{2}}, \sigma^2\left(2 - \frac{\pi}{2}\right)\right).$$

We could then calculate $KL(q(X \mid b, 0)\|q(X \mid 0, 0))$ by

$$\frac{1 + \frac{KL(q(X|b,0)\|q(X|0,0))}{KL(q(Z_{1,1})\|q(Z_2))}}{1 + \frac{KL(q(X|b,0)\|q(X|0,0))}{KL(q(Z_{1,2})\|q(Z_2))}} = \frac{\log \hat{P}_F(Z_{1,1})}{\log \hat{P}_F(Z_{1,2})},$$

where $Z_{1,1} \sim \mathcal{N}(\hat{\sigma}_{11}\sqrt{\frac{\pi}{2}}, \hat{\sigma}_{11}^2\left(2 - \frac{\pi}{2}\right))$ and $Z_{1,2} \sim \mathcal{N}(\hat{\sigma}_{12}\sqrt{\frac{\pi}{2}}, \hat{\sigma}_{12}^2\left(2 - \frac{\pi}{2}\right))$. $\hat{\sigma}_{11}$ and $\hat{\sigma}_{12}$ are two small positive values, namely the scale parameters of the Rayleigh distributions $Z_{1,1}$ and $Z_{1,2}$. In this simulation, we take $\hat{\sigma}_{11} = 0.25$ and $\hat{\sigma}_{12} = 0.35$ respectively.

The simulation results on the additive noise following Rayleigh distribution also validate our proposed theoretical performance upper bound.

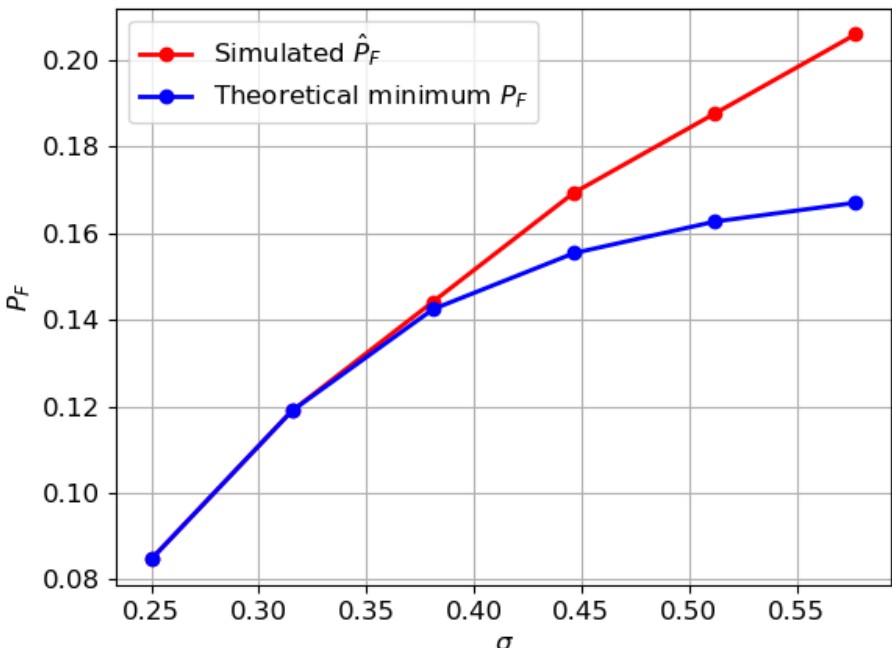

Figure 3: Simulated $\hat{P}_F$ and the proposed theoretical minimum $P_F$ as a function of the standard deviation of the additive noise.

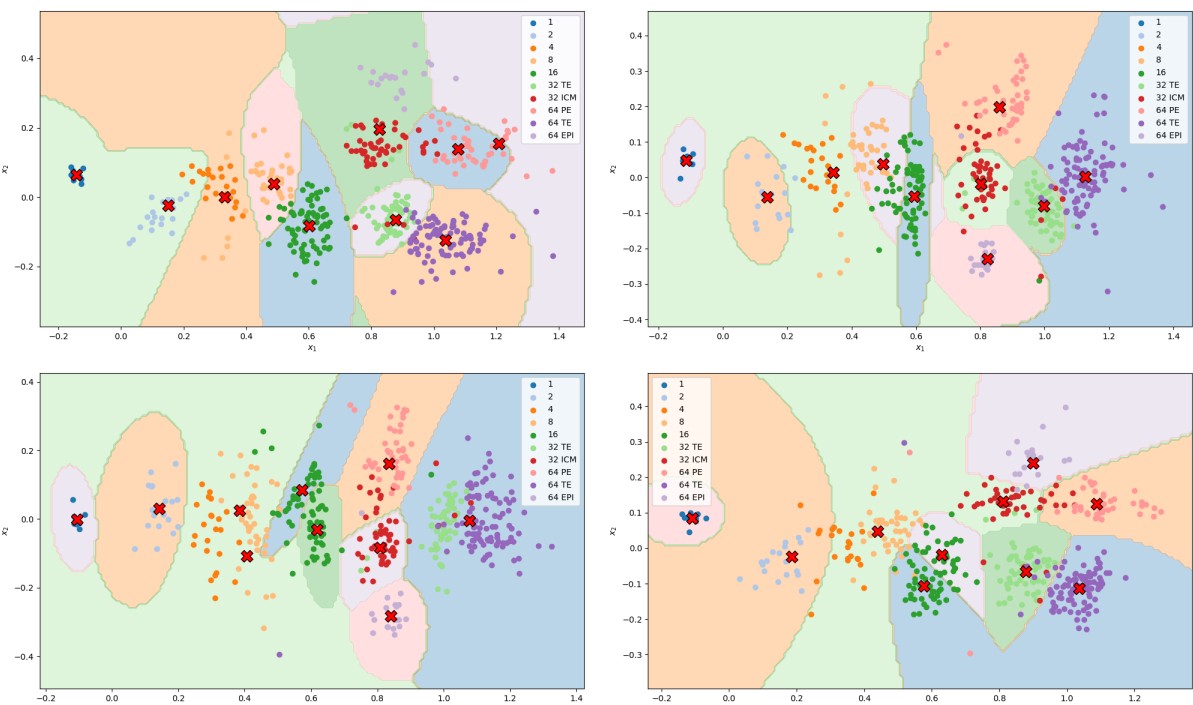

Figure 4: Simulation results of clustering the noise-corrupted single-cell qPCR data using a GMM, with dimension reduction performed by BGPLVM. The upper left, upper right, lower left, lower right subfigures correspond to the data with Rayleigh noise at scale parameters of $\sigma = 0.1, 0.2, 0.3, 0.4$ respectively.

## 5 Conclusions

The probabilistic models have been widely used in learning problems where uncertainty is an important issue, e.g. the data are corrupted with noises. However, in spite of the promising results of these algorithms in numerous real-world applications, their performance in learning tasks with noise-corrupted dataset has not yet been quantitatively analyzed in the literature. In this paper, we propose to analyze the upper bound of performance of BGPLVM for the clustering task, enabled by the transparent probabilistic inference structure of this probabilistic model. The performance upper bound has merit in the clustering tasks by BGPLVM, but can also be generalized to a broader class of probabilistic models. The numerical simulation results for the additive Gaussian noises and Rayleigh noises are given to validate the proposed theoretical performance bound. The results of this paper reveal the superiority of the probabilistic models in safe learning tasks where safety and quantifiable error are strongly desired.

### Broader Impact Statement

Although the performance upper bound is given for the clustering task using GPLVM, it can be extended to other probabilistic models for other learning tasks. It provides a quantitative performance deterioration of a probabilistic model with noise, which is quite significant for learning problems with noises however requiring a desired performance guarantee.

## A Function $l(\cdot)$ and the approximation details

In this appendix, we begin by providing an illustrative example of the function $l(\cdot)$, which meets the conditions $0 \leq l(x) \leq x$ and $\frac{l(x_1)}{x_1} \leq \frac{l(x_2)}{x_2}$ for $0 \leq x_1 \leq x_2$. This illustration is shown in Figure 5. The closer the red curve is to the blue one, the clustering(detection) algorithm has better performance.

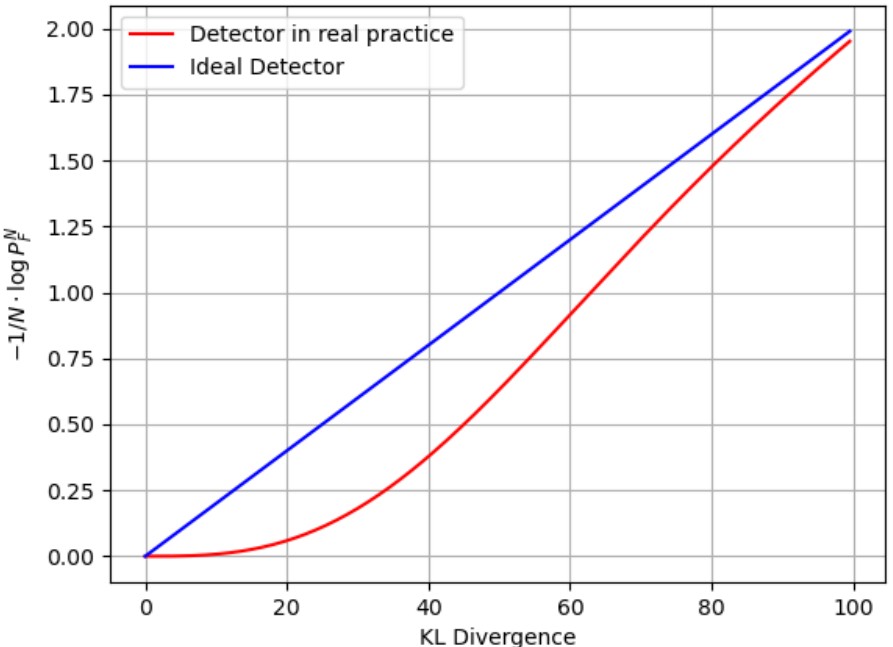

Figure 5: The function $-\frac{1}{N} \cdot \log P_F^N$ for the ideal detector and a practical detector as a function of the Kullback-Leibler distance, with a large $N$.

Next, we provide a detailed explanation of the approximation in equation 12. Let us define:

$$KL^{\oplus}(\sigma) := KL\left(q(\sigma^2)\|q(Z_2)\right) + KL(q(X \mid b, 0)\|q(X \mid 0, 0))$$

Using equation 7, we obtain:

$$\frac{\log \hat{P}_F(\hat{\sigma}_{11})}{\log \hat{P}_F(\hat{\sigma}_{12})} = \frac{l\left(KL^{\oplus}(\hat{\sigma}_{11})\right)}{l\left(KL^{\oplus}(\hat{\sigma}_{12})\right)},$$

where $\hat{\sigma}_{11}^2$ represents the inherent noise in the dataset. This means that $\hat{P}_F(\hat{\sigma}_{11})$ is calculated without adding any additional noise to the original dataset. On the other hand, $\hat{\sigma}_{12}^2 = \hat{\sigma}_{11}^2 + 0.1^2$, where $0.1^2$ is the variance of the added noise, which is independent of the inherent noise, leading to the direct summation of their variances.

We observe in Figure 5 that when selecting a relatively small pair of $\hat{\sigma}_{11}^2$ and $\hat{\sigma}_{12}^2$, the slopes are approximately equal

$$\frac{l\left(KL^{\oplus}(\hat{\sigma}_{11})\right)}{KL^{\oplus}(\hat{\sigma}_{11})} \approx \frac{l\left(KL^{\oplus}(\hat{\sigma}_{12})\right)}{KL^{\oplus}(\hat{\sigma}_{12})},$$

resulting in

$$\frac{\log \hat{P}_F(\hat{\sigma}_{11})}{\log \hat{P}_F(\hat{\sigma}_{12})} = \frac{l\left(KL^{\oplus}(\hat{\sigma}_{11})\right)}{l\left(KL^{\oplus}(\hat{\sigma}_{12})\right)} \approx \frac{KL^{\oplus}(\hat{\sigma}_{11})}{KL^{\oplus}(\hat{\sigma}_{12})}.$$

This leads to the approximation given in equation 12.

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
