# OpenReview forum: "Gaussian Process Latent Variable Model: A Performance Upper Bound with Noise-Corrupted Data"
_TMLR — Withdrawn by Authors_

### Review · Reviewer_ngDL · 2024-10-31

**Summary Of Contributions:**

This paper considers a Bayesian Gaussian Latent Variable Model to perform probabilistic non-linear dimensionality reduction on noise-corrupted data. Specifically, additive Gaussian noise is considered. After the dimensionality reduction, a clustering task is analyzed. The paper aims at deriving a upper bound on the clustering performance. The authors evaluate their method on one small-scale real-world data set using a Gaussian Mixture Model.

**Audience:**

Yes

**Broader Impact Concerns:**

I do not identify any ethical concerns.

**Claims And Evidence:**

No

**Requested Changes:**

I believe that the paper needs a major revision to improve clarity and a convincing experimental evaluation. Addressing the aforementioned weaknesses, questions, and minor comments would be a start.

**Strengths And Weaknesses:**

### Strengths

1. The idea of deriving a upper bound on the clustering performance is appealing.
2. The paper contains a good mix of theory and practice.

### Weaknesses

1. The paper lacks clarity. Many variables/symbols are undefined and have to be guessed or inferred from context. The same is true for details of the experimental evaluation. See my list of questions and minor comments below.
2. Section 2 mentions that it follows the definition of McHutchon & Rasmussen (2011) which is not true. It follows Titsias & Lawrence (2010). Moreover, it does not only follow it, the majority of Section 2 is directly copied from Sections 2 and 3 from Titsias & Lawrence (2010) with minor word adjustments.
3. The paper only considers one small data set and one clustering method. Details of the experimental evaluation are missing and the effect of certain choices not evaluated and thus unclear. See especially Question 13 below.
4. The title is more general then what the paper actually does. It only considers a clustering scenario.
5. I have problems believing that the statement *"upper bound represents the best performance achievable by any clustering algorithm, regardless of the specific algorithm used, with BGPLVM serving as the means of dimensionality reduction"* made in the abstract is true. Why should it hold for any clustering algorithm? I could define a very bad one. If the bound holds it might be useless. Furthermore, how can it be independent on how I do the dimensionality reduction, i.e., my choice of kernel?

### Questions

#### General
1. Why is it important to perform the non-linear dimensionality reduction using a probabilistic/Bayesian model? As far as I understood, the clustering is done on the $\mu_n$ values (Equation (3)) and the covariances $S_n$ are not used.
2. Consider the sentence *"[...] the true latent variables $X$ are not affected by the noise $Z$, making them independent"* from the proof of Theorem 2. If it was true, then what is the point of the paper? It suggests that the noise does not effect the true latent variables on which the clustering algorithm acts.
3. What role does clustering actually play here? Based on Section 3, couldn't one equivalently also consider a classification scenario?
4. Can you elaborate on the end of page 7 that states *"Since our training data is labeled [...]"*. Is this just to make the evaluation of this controlled setting easier? I understand that point but it is not practical as in a clustering scenario, there is no access to labeled data.
5. If Lemma 3 provides a solution that holds in the limit (number of observations $N\to\infty$) and Monte Carlo is used (Section 4), then how is the bound analytic as claimed in the abstract? What does analytic refer to?

#### Specific

6. Why is there no independence assumption in Theorem 2? It is needed to use Lemma 1. How does the first line in the proof of Theorem 2 follow from Lemma 1?
7. If $X$ and $Z$ are independent and they factorize, what role does $q(X|Z)$ (end of page 5) play? According to the independence, $Z$ should not influence $X$.
8. Why is Corollary 2.1 not restated as Lemma 1? I cannot find Corollary 2.1 in Polyanskiy & Wu (2014).
9. Within the first four pages, everything is about additive noise. Section 3.2 then also introduces a bias. Why is this needed (and not mentioned before)?
10. Section 3.1: What is $M$? Are the subsets randomly chosen? Of approximately the same size? Is it the number of true categories? And what exactly is $\mathbf{C}$? Why is it bold? The sentence suggests that $\mathbf{C}$ contains the true labels. But if I have $M$ classes, then $\mathbf{C}_i$ just contains a number of the same labels.
11. If $M$ denotes the number of classes, why is the number of samples drawn from the data set within Monte Carlo (Section 4) dependent on the number of classes? Is the drawing uniform? With or without replacement? What are the two subsets for? What is $\mathcal{M}_{1,i}$ and how is it related to the bias?
12. What are the specifics of the data set? Based on the text I assume 437 data points in 48 dimensions from 10 classes which are embedded in a two-dimensional space. Isn't this $N$ too small for Lemma 3?
13. The experiments lack clarity. For example in Figure 1: Are those all data points? Do they come from 10 classes? What do the red crosses indicate? I assume cluster centers. What are the colors of the data points and underlying areas representing? Which $\beta$ was used? Which kernel was used? Does the kernel have hyerparameters? What is the latent dimensionality $Q$? Is it two? Are the $\mu_N$ values clustered? Are the $S_N$ values used at all? Are the number of clusters equivalent with the number of classes? What about the bias $b$? Are the added noise levels relevant, i.e., are they in a meaningful scale? Was the experiment only executed once? Are the results robust when the previously mentioned values and design choices are altered?


### Minor Comments
- Consistency: "datasets" vs. "data sets"; "Gaussian Process" vs. "Gaussian process"; "i.e.," vs. "i.e."; "Type I error" vs. "type-I error"; "Chernoff-Stein lemma" vs. "Chernoff-Stein Lemma"
- Wrong citation style: "author (year)" instead of "(author, year)" should be used in the first line of Section 2. Same in the beginning of page 4.
- Some sentences start with "And", even in sequence.
- Text is missing between Equation (1) and Equation (2).
- The meaning and space of some variables is never mentioned and has to be either guessed or inferred from context. Examples: $\beta$ (Equation (2)), $F_{N}$, $q_{N}$, $\tilde F_{N}$, $M$, $P_{F}^{i}$, $\mathcal{Y}$, $\mathcal{P}(\mathcal{Y})$, $\mathcal A_{N}$, $\mathcal A_{N}^{c}$, $\overset{\cdot}{=}$, $P_{M}^{N}$, $P_{F}^{N}$, $P_{F}^{N_{*}}$, $\mathcal M_{1,i}$, $\sigma_{2}^{2}$, etc.
- "Equation (x)" instead of "equation x"
- Math is always a part of a sentence and, thus, should have proper punctuation. An equation should not appear without text.
- *"The marginal likelihood of the data $p(y)=$[...]"* is not a proper sentence. (top of page 3)
- Please avoid starting a sentence with a variable.
- Why is the left-hand side of an equation often on top of the first equal sign instead of left of it? See, for example, $F(q)$ on page 3.
- There is a typo on page 3, right before Lemma 1: It is mentioned that the second term of the equation with $F(q)$ is the KL divergence between the variational posterior distribution $q(X)$ and the prior distribution $p(X,Y)$. The prior is just $p(X)$.
- In the middle of page 3, *"In the following theorem [...]"* is stated but no theorem follows. First, there is Lemma 1.
- For Lemma 1, non-published lecture notes (that I cannot access) are cited. Moreover, there is no text at all.
- Theorem 2: The abbreviation KLD was never introduced. The symbols $F_N(q_N)$ and $q_N(X,Z)$ appear which have not been introduced. What does $N$ stand for? The number of observations (Section 2)? My guess is noise but it should be clearly stated.
- Lemma 3 (Chernoff-Stein Lemma): After checking Cover (1999), $\epsilon$ should be in $(0,\frac12)$. Some symbols/variables used in this lemma are never introduced.
- There is a typo in the middle of page 5: The dimension of a data sample is $D$ not $N$.
- It is only clear from context that the 0 in $q(X|0,Z_2)$ refers to the bias. This should be more clear.
- Lemma 4: The role and choice of the function $l(\cdot)$ is unclear.
- Lemma 5 and Theorem 6: Text is missing.
- Beginning of page 8: Why is $Z_1$ drawn from many different distributions? I assume there is a typo.
- Figures 2, 3, and 5: The figures should be smaller in size.

---

### Review · Reviewer_yb9r · 2024-11-02

**Summary Of Contributions:**

The paper proposes to use Chernoff Stein Lemma to provide an analytic upper bound on quantitative clustering.

**Audience:**

Yes

**Claims And Evidence:**

Yes

**Requested Changes:**

Please see the strengths/weaknesses section for questions that should be addressed before re-submitting.

**Strengths And Weaknesses:**

The problem itself is interesting, and the relationship could be useful.

Weaknesses:

The paper needs a serious re-write. I stopped reading at Section 3.3 because there were too many things that make no sense. I did not go through the numerical experiments because the front matter has too many clarifications needed.

Notations are confusing and there is a bunch of overloading that makes no sense. Is \underaccent{\bar}{Y} used and defined in Theorem 2 a random variable or is it fixed? If it is fixed data, Lemma 1 is meaningless since it is defined for X,Y,Z random variables. On the other hand, if it is random variable, then P(Y = \underaccent{\bar}{Y}) does not make sense.

I think the 1st sentence of proof of Theorem 2 is incorrect. Please double-check and clarify: should it be p_N(Y|X) = P(Y= underbarY|X) P(underbarY +Z). Why is Z part of the LL function F that needs to be maximized?

I am not sure I understand why Theorem 2 is interesting at all. It should be obvious that adding independent gaussian noise to the observations when underlying model is a GP would not change the model as the noise is captured in the latent representation (which is what theorem 2 proves).

“We could regard clustering algorithm as a detector” – I am not sure what this means. What is a a detector algorithm? It takes as input the data point and outputs the cluster index? How is a false alarm for Y* \notin hatC_i the same as \hat{C}_i \neq  C_i when C_i itself has not be used anywhere in the modified multihypothesis testing?

The inequality when the limit is removed from Chernoff-Stein lemma is not clear to me (pg 5 below Lemma 3).

Is “false alarm” type 1 error (as used in section 3.1) or type 2 error (as used in sec 3.2) ? The relationship with the clustering/detection is still not obvious to me.

The function l(.) is a detector algorithm as used in Proof of lemma 4. What is its input and output?

How are the different q with some bias b added surrogates for different distributions P_0 and P_1 (in Lemma 3) ? I don’t understand the application of Lemma 3 using different q (which are approximations learnt through the GP algorithm, not actual sampling distributions). Where did the bias suddenly turn up from?

Minor:

Please use different variables for Lemma 1 or clarify: With X,Y being just introduced, Lemma 1 is confusing because Y = X + Z does not make sense since the introduced X,Y have different dimensions. Lemma 1 as it is used in Theorem 2 is

F_N(q_N) is not defined before it is used in Theorem 2.

Is N dimension or number of observations in Lemma 3?

---

### Review · Reviewer_VLxy · 2024-11-09

**Summary Of Contributions:**

This paper concerns the performance classification algorithms using BGPLVM as a dimensionality reduction technique. BGPLVM is first reviewed, and discussed in the setting where noisy data is observed. A classification problem is then presented and reformulated as a multiple hypothesis testing problem. The Chernoff-Stein Lemma is introduced, and a result is presented to quantify the impact of noise on the false alarm rate in the classification problem. Finally, some numerical results are presented examining the theoretical bounds and the empirical type-II error.

**Audience:**

No

**Claims And Evidence:**

No

**Requested Changes:**

The authors should significantly clarify their framework and results, if correct. BGPLVM should also be introduced in the authors' own words.

# References

Dalmaijer, E. S., Nord, C. L., & Astle, D. E. (2022). Statistical power for cluster analysis. _BMC bioinformatics_, _23_(1), 205.

Kay, S. M. (1998). Fundamentals of Statistical Signal Processing: Detection theory. Prentice-Hall.

McHutchon, A., & Rasmussen, C. (2011). Gaussian process training with input noise. _Advances in Neural Information Processing Systems, 24.

Titsias, M., & Lawrence, N. D. (2010). Bayesian Gaussian process latent variable model. In _Proceedings of the Thirteenth International Conference on Artificial Intelligence and Statistics_ (pp. 844-851).

**Strengths And Weaknesses:**

## Strengths
- In general, deriving information-theoretic performance bounds on how noisy data impacts downstream tasks is an interesting topic.

## Weaknesses

I overall find this paper quite confusing to follow, and was unable to understand many of its assertions enough to verify their results. After reading carefully, I have several doubts as to the correctness of the authors' claims. Even if the results can be fixed, I believe substantial rewriting would be necessary.

**The Title is Too Vague**
I think the title may be a bit too broad, and potentially misleading. While classification is certainly an application of GPLVM, it is not the only one (and I'm not convinced it's the main one), and the title should be clear that classification is the problem considered.

**On the Formulation of the Clustering Problem**
I don't think I understand the formulation of clustering as a hypothesis test. For one, the authors assume that the set of potential labels $C$ is known, and the task is to determine whether an observation $Y^*$ comes from cluster $C_i$. This seems like a classification task to me, not a clustering problem.

Moreover, the authors write "if we further assume that each data sample has a unique category, only one alternative hypothesis $H_1^i$ will be accepted." I don't see why this is true. In general, multiple hypothesis testing is not equivalent to simultaneous hypothesis testing (see, e.g., Section 3.8 of Kay (1998) for discussion).

Finally, it is stated that "we are interested in the error (misclustering) rate (Delmaijer et al., 2022)." I can find no reference to the term "misclustering rate" in Delmaijer et al. (2022), and the hypothesis test considered in their paper seems fundamentally different than the one considered here. For example, from page 5 of Delmaijer et al. (2022), "analogously, in cluster analysis the aim is to establish whether a dataset contains subgroups, or comprises a single group. We thus defne statistical power in cluster analysis as the probability of correctly detecting that subgroups are present."

**On the Connection to the Chernoff-Stein Lemma**
The Chernoff-Stein Lemma concerns a hypothesis test between two alternative hypotheses $H_1 \colon Q = P_1$ and $H_2 \colon Q = P_2$ using i.i.d. samples from $Q$, relating the exponent of the type-II error to the KL divergence between $P_1$ and $P_2$. I don't understand what $P_1$ and $P_2$ are here.

It seems like Lemma 4 directly applies the Chernoff-Steim Lemma to $P_1 = q(X | b, Z_1)$ and $P_2 = q(X | 0, Z_2)$, but it's not clear to me what this actually means as a hypothesis test, or how it's related to the clustering problem. I also don't know what $N$ represents in this case, or what counts as an observation. Further, the Chernoff-Stein Lemma is an asymptotic result, and cannot be applied directly to finite samples.

**Significant Overlap with Titsias & Lawrence (2010)**

Section 2, which provides an overview of the BGPLVM, has a significant textual overlap with Titsias & Lawrence (2010). Consider the following text from Titsias & Lawrence (2010):
>**Let $Y \in \mathbb{R}^{N\times D}$ be the** observed data **where $N$ is the number of observations and $D$ the dimensionality of each data vector.** These **data are associated with latent variables $X \in \mathbb{R}^{N \times Q}$** where, for the purpose of doing **dimensionality reduction, $Q \ll D$.** The GP-LVM (Lawrence, 2005) defines a forward (or **generative) mapping from the latent space to observation space that is governed by Gaussian processes. If the GPs are taken to be independent across the features then the likelihood function is written as (Eq.1)**, where $y_d$ represents the $d$th column of $Y$ and **(Eq. 2)**.

and this submission:

> Following the definition of BGPLVM in (McHutchon & Rasmussen, 2011), **let $Y \in \mathbb{R}^{N\times D}$ be the** observation data set **where $N$ is the number of observations and $D$ the dimensionality of each data vector.** And we assume the **data are associated with latent variables $X \in \mathbb{R}^{N \times Q}$**. And as the BGPLVM is used for **dimensionality reduction,** we have **$Q \ll D$.** The BGPLVM **defines a generative mapping from the latent space to observation space** which **is governed by Gaussian processes.** **If the GPs are taken to be independent across the features then the likelihood function is written as (Eq.1) (Eq. 2).**

While I understand that it can be difficult to rephrase technical content, this continues for a full page of content, and seems a pretty extreme reuse of text. McHutchon & Rasmussion (2011) is also unrelated to the BGPLVM (or GPLVMs in general); I assume the authors meant to cite Titsias & Lawrence (2010) here.

**Other Various Inaccuracies**
There are various other smaller inaccuracies, for example:
- Page 1 states "In BGPLVM, the latent variables are not integrated out but are instead optimized [...]", but the entire point of BGPLVM is to variationally integrate out the latent variables instead of just optimizing them.
- Lemma 4 assumes that the performance of an arbitrary detector improves monotonically with the KL divergence; I do not believe this is true for arbitrary detectors.

Some other results were not carefully checked (including the majority of the numerical section) since I could not understand the rest of the paper, and I suspect more results have errors. But the above is, in my opinion, already enough such that a minor revision would not be sufficient.

---

### Note · Authors · 2024-11-10

**Comment:**

As one reviewer proposed, multiple hypothesis testing is not equivalent to simultaneous hypothesis testing, which invalidates some results of the paper. We decide to withdraw the paper.

**Withdrawal Confirmation:**

I have read and agree with the venue's withdrawal policy on behalf of myself and my co-authors.